# Impact of the 13-Valent Pneumococcal Conjugate Vaccine on Severe Invasive Disease Caused by Serotype 3 Streptococcus Pneumoniae in Italian Children

**DOI:** 10.3390/vaccines7040128

**Published:** 2019-09-24

**Authors:** Lorenzo Lodi, Silvia Ricci, Francesco Nieddu, Maria Moriondo, Francesca Lippi, Clementina Canessa, Giusi Mangone, Martina Cortimiglia, Arianna Casini, Ersilia Lucenteforte, Giuseppe Indolfi, Massimo Resti, Chiara Azzari

**Affiliations:** 1Department of Health Sciences, University of Florence and Meyer Children’s University Hospital, viale Pieraccini 24, 50139 Florence, Italy; 2Department of Clinical and Experimental Medicine, University of Pisa, via Savi 10, 56126 Pisa, Italy; 3Department of Pediatrics, Meyer Children’s University Hospital, viale Pieraccini 24, 50139 Florence, Italy

**Keywords:** serotype 3, streptococcus pneumoniae, PCV13, vaccination impact, sepsis

## Abstract

The effectiveness and impact of the 13-valent pneumococcal conjugate vaccine (PCV13) against invasive pneumococcal diseases (IPD) due to serotype 3 (ser3) has been questioned. However, the impact of PCV13 on different clinical presentations of ser3-IPD has not been studied so far. The impact of PCV13 on different clinical presentations of ser3-IPD in a population of Italian children aged 0–8 years was evaluated, comparing pre- and post-PCV13 introduction period. Real-time polymerase chain reaction (PCR) was used for the diagnosis and serotyping of IPD. During the observation period (1 January 2006–1 August 2018), ser3 was detected in 60/284 (21.1%) children under 8 with serotyped IPD. The incidence of sepsis and meningitis was 0.24 per 1,000,000 person-years (p-y) in pre-PCV13 and 0.02 per 1,000,000 p-y in post-PCV13. No cases occurred in vaccinated children. In the post-PCV13 period, case reduction was 13% for all ser3 IPD and 92% for sepsis and meningitis. Vaccination impact may be underestimated due to significant improvement in pneumococcal surveillance in post-PCVC13. Our data suggest a significant impact of PCV13 on meningitis and sepsis due to ser3 and a lower impact against pneumonia. While waiting for increasingly effective anti-pneumococcal vaccines, PCV13, which guarantees protection against the most severe clinical presentations of ser3-IPD, is currently the most effective prevention option available.

## 1. Introduction

The effectiveness and impact of the 13-valent pneumococcal conjugate vaccine (PCV13) against serotype 3 *Streptococcus pneumoniae* (ser3) has been repeatedly questioned [1,2,3,4,5]. Ser3 pneumococcus has established itself as one of the leading causes of invasive pneumococcal disease (IPD), even in the post-PCV13 introduction period (post-PCV13) [6], and has been associated with complicated and severe infections in children [5,7,8,9]. This aggressive behavior can be firstly attributed to microbiological features of the pathogen-like thick polysaccharide capsule [10]. Moreover, in this strain, capsular polysaccharide (CPS) is produced by a synthase mechanism that does not covalently link it to peptidoglycan, so allows the release of free CPS during growth [11]. Free CPS can saturate anti-ser3 specific antibodies, reducing opsonophagocytic killing and neutralization [10,11]. Currently, there is open debate regarding the effectiveness of PCV13 against ser3 pneumococcus, with some post-licensure studies showing non-significant [2], non-evaluable [3], or extremely reduced [4] effectiveness, while others [12,13,14] report significant protection. There is an essential need and an open call in the scientific community for impact studies on PCV13 against ser3 IPD [14,15]. The intent of the present paper is to give a contribution to that call. It is assumed [16] that different clinical presentations of IPD require different amounts of antibodies, with the lowest levels needed to prevent meningitis and sepsis and the highest for pneumonia and carriage state. However, previous reports did not evaluate whether PCV13 may have a different impact on different clinical presentations. Therefore, the aim of this study is to evaluate the impact of PCV13 on different clinical presentations of IPD caused by ser3 in a population of Italian children under 8 years of age and to evaluate the severity of clinical manifestation in unvaccinated and vaccinated children.

## 2. Materials and Methods

### 2.1. Study Design

This observational study was conducted retrospectively from January 2006 to August 2018 and evaluated all children admitted to Italian hospitals with the diagnosis of IPD based on data from the national Molecular Surveillance Register (MSR). As previously described [17], the MSR was started at the Immunology and Infectious Diseases Laboratory, Meyer Children’s Hospital, Florence, Italy, (hereinafter “central lab”) in 2006 and it was further expanded thanks to dedicated funds from the Italian Center for Disease Control (CCM). Participation of all Italian pediatric units and hospitals was encouraged but it was not mandatory. Patients were included in the MSR only if at least one biological sample (peripheral blood, pleural fluid, and cerebrospinal fluid) had been tested using real-time polymerase chain reaction (RT-PCR) at the central lab; clinical and laboratory data were recorded and the results of culture tests were also recorded when available. Clinical information was obtained by completing a standardized questionnaire, including sex, age, and vaccination status at the moment of sample collection. Samples from patients with severe concomitant disease, (i.e., cystic fibrosis, immunodeficiency, neurological impairment) or suspicion of nosocomial infection were tested and included in the MSR but excluded from the present study.

The study was approved by the local review board. No specific approval by the regional pediatric ethic committee was required for this study, since all samples were collected as part of routine clinical practice and data were analyzed retrospectively and anonymously.

### 2.2. Case Definition

In agreement with the Center for Disease Control and Prevention (CDC; Atlanta, U.S.A.), IPD was defined as a case of infection which met laboratory criteria for the identification (RT-PCR) or the isolation (culture) of *Streptococcus pneumoniae* from a normally sterile body site [18]. As previously described [19], pneumonia was suspected on the basis of clinical signs and confirmed by chest radiographic examination or computed tomography (CT) [20]. Pneumonia was classified as community-acquired (CAP) when the infection was transmitted in a non-nosocomial setting, as distinguished from hospital-acquired pneumonias [21], which were not included in this study.

Complicated CAP was defined as the presence of parapneumonic effusion, empyema and/or massive/atelectasis or necrotizing pneumonia [19,22,23]. Sepsis [24] and meningitis [25] were grouped together in this study and considered “severe systemic infections”. Regardless of the initial or concomitant presence of an infectious localization in specific body sites (e.g., pneumonia or otomastoiditis), when the patient presented the clinical criteria of sepsis or meningitis the case was recorded as “sepsis” or “meningitis” and thus grouped among the “severe systemic infections”. In this way it was possible to group all cases that had a severe course within the group of “severe systemic infections”. Patients affected by IPD with “extremely severe clinical course” were identified as those within the group of “severe systemic infections”, plus those who, regardless of infection type, needed admission to a pediatric intensive care unit (PICU) for life support. Patients were divided into two groups based on whether they were born before or after PCV13 introduction in the Italian vaccination schedule. Further sub-grouping was based on effective vaccination status at the moment they developed ser3 IPD. The exact patient subdivision is detailed in Section 3.1.

### 2.3. Laboratory Methods

Laboratory confirmation was obtained by RT-PCR and/or culture methods, as previously described [7,19,26]. Whole blood and/or other biological fluids were collected from all patients as soon as possible after hospital admission for culture and/or molecular tests. Samples for molecular tests were sent by the participant centers to the central lab at room temperature using an overnight freepost carrier; molecular tests were performed within 2 h of delivery. Stored isolates were also accepted. All samples were tested with RT-PCR for the *lytA* gene, as previously described [19]. Isolates already classified as *S. pneumoniae* by cultural methods were also sampled and analyzed for the *lytA* gene in order to confirm the diagnosis. A sample was considered negative if there was no increase in fluorescent signal before RT-PCR cycle 40. An etiological diagnosis was made if RT-PCR and/or culture was positive in blood or pleural fluid, as previously described [19].

All samples were serotyped using RT-PCR. For both diagnosis and serotyping by RT-PCR, 200 μL of available biological fluids or lyophilized isolates were used [27]. For serotyping, 38 primer/probe sets targeting different regions of the *cpsA* gene were used, specific for 33 different serotypes. Twenty-nine primer–probe sets were previously published by our group [19,28,29]. The sequences of the nine additional primer–probe sets are available upon request. If no increase in fluorescent signal was observed after 40 cycles for any of the serotype-specific primer/probe sets, in spite of a positive result with both RT-PCR (*lytA* gene) and end-point PCR (*cpsA* gene) [19], the sample was considered non-typeable with the serotype-specific primers in RT-PCR.

### 2.4. Evaluation of IPD Incidence and Impact of Vaccination

PCV13 was included in the Italian vaccination schedule in the last quarter of 2010, so ser3 IPD cases were divided according to whether the patients were born before or after 1 January 2011—pre- or post-PCV13. The vaccine is administered with three doses in the first year of life (at the third month, at the fifth month, and at the 11th–13th month). No catch-up strategy has been implemented in Italy (not even at regional level) in order to guarantee vaccination coverage with PCV13 for children born before 2011, so all children born before 2011 are considered as unvaccinated against ser3. Ministerial data, updated as of 31 December 2017, show a national vaccination coverage for PCV13 of 90.90% in children of 24 months of age, 90.35% in those 36 months-old, and 89.41% in those 48 months-old [30,31,32]. Data on coverage are given in detail in Appendix A. During the pre-PCV13 period, the Italian Ministry of Health had not implemented a routine data collection on vaccination coverage for *Streptococcus pneumoniae*. The incidence of ser3 IPD was calculated using a group-specific person–year denominator, which took into account the length of the observation period and the overall Italian pediatric population 0–8 years old, based on data from the National Institute of Statistics [33]. Details on denominators and source are given in Appendix A. Since participation in MSR is dependent on voluntary bases, the incidence of ser3 IPD cases calculated using the whole Italian pediatric population under 8 does not reflect the true incidence of disease in Italy, but allows for an evaluation of vaccination impact.

Vaccine impact was estimated for vaccine eligible ser3 cases (born after 1 January 2011). Cases that occurred in post-PCV13 in vaccine eligible children were compared with cases that occurred in the pre-vaccine period. Crude incidence was used for the pre-vaccine period (reference population) and an age-standardized rate was used for the post-vaccine period (study population). Vaccination impact was calculated for ser3 overall IPD cases and then specifically for ser3 pneumonias and for ser3 severe systemic infections.

The attention paid by clinicians to surveillance increased over time, as shown by the ministerial data reported in Appendix A; this point was thus evaluated in order to ascertain the potential bias associated with an increased number of diagnoses achieved in more recent years. Further stratification of patients born after 2011 was based on data from MSR regarding vaccination status at the moment of sample collection. A patient was considered vaccinated if they had received an appropriate number of doses for their age.

### 2.5. Statistical Analysis

Data were processed with the freely available “epitools” R package [34]; *p* < 0.05 was considered to be statistically significant. Results were expressed as the mean and standard deviations (SD) or as the median and interquartile range (IQR), as appropriate. The Pearson chi-square test or Fisher’s exact test were used to assess group differences in categorical variables. Incidence rates were calculated as overall crude incidences, age-specific incidences, and age standardized rates (ASR). ASR were calculated through direct age standardization, considering the population of children born before PCV13 introduction (pre-PCV13) as the reference population and the population of children born after PCV13 introduction (post-PCV13) as the study population. The standardized incidence ratio (SIR) was calculated as the ratio between the sum of observed cases of post-PCV13 and the sum of expected cases of post-PCV13. The same methodology of direct and indirect age standardization was applied when analyzing incidence and risk for the subgroup of patients born after PCV13 introduction and vaccinated. The odds ratio and 95% confidence intervals (CIs) were calculated, when possible. Confidence intervals were calculated using the exact methods and Agresti–Coull method, depending on the type of variable analyzed. For continuous variables, the Student’s t test was used.

## 3. Results

### 3.1. Demographic Data of Study Populations and IPD Serotype Distribution

From 1 January 2006 to 1 August 2018, we recorded 386 IPD in patients under 18 years of age; of these, 326 (84.4%) were younger than 8 years of age at the moment of sample collection. Pneumococcal serotype identification was possible in 340/386 patients under 18 (88.0%) and in 284/326 patients under 8 (87.1%).

Ser3 was detected in 60/284 children under 8 years of age (21.1%). Vaccination status for PCV13 at the moment of sample collection was unknown in one of them, who was excluded from the analysis. Fifty-nine patients with ser3 IPD (34 males, 57.6%; mean age 3.30 ± 1.67 years; median age 3.07 years; IQR 2.08–4.33 years) were therefore evaluated. Data from culture-based tests were available only in 29% of patients with ser3 IPD (17/59) and only 12% of the total (7/59) had a positive result with the isolation of *Streptococcus pneumoniae*. Patients were divided into three groups (Figure 1): group A, patients born before 2011 and therefore not vaccinated with PCV13 (38/59); group B, patients born after 2011 (21/59); and group B1 (sub-group of B), patients born after 2011 and vaccinated with PCV13 (16/59) at the moment of sample collection. The age distribution was different between A and B, so the incidence rates were calculated crude and then adjusted for age. In group A, the vaccination status for PCV7 was known for 27/38 children; of these, 85.2% had received a complete vaccination series.

Changes in IPD serotype distribution were recorded in patients born before and after PCV13 introduction. Serotype-specific data regarding distribution in groups A, B, and B1 are detailed in Appendix A. Serotype distribution data are presented in order to provide the reader with a better epidemiological contextualization and describe which were the predominant serotypes in the IPD of Italian children pre- and post-PCV13. Extensive presentation and discussion of serotype-specific data is beyond the purpose of this paper, the intent of which is a specific analysis of PCV13 impact on IPD by ser3.

### 3.2. Impact of PCV13 on ser3 IPD Incidence Rates in a Cohort of Italian Children under 8

Age-specific ser3 IPDs incidence rates in children born before 2011, born after 2011, and in children born after 2011 and vaccinated are reported in Figure 2A. Ser3 IPD’s crude incidence rate was 0.99 per 1,000,000 person-years (p-y) in children under 8 born before 2011 (95% CI: 0.96–1.02). Age standardized incidence rates (ASR) were 0.87 per 1,000,000 p-y in children under 8 born after 2011 (95% CI: 0.73–1.01) and 0.62 per 1,000,000 p-y in the subgroup of children born after 2011 and vaccinated with PCV13 (95% CI: 0.38–0.86) (Table 1 and Figure 3). A punctual rates comparison showed a decrease of 0.12 and 0.37 cases per 1,000,000 p-y, respectively, in children born after 2011 and in the subgroup of children vaccinated with PCV13 when compared to unvaccinated children born before 2011. The standardized incidence ratio (SIR) for vaccinated children only, compared to unvaccinated children born before 2011, was 0.75 (95% CI: 0.54–0.96).

PCV13 impact on ser3 IPD was 0.13 (95% CI: −0.59–0.51).

As previously described [7], the number of samples received by the central lab for the surveillance of pediatric invasive bacterial diseases increased by 102% between the pre-PCV13 and post-PCV13 periods (from a mean annual values of x samples and a mean annual value of y samples), in accordance with data obtained by the Italian National Institute of Health (see Appendix A).

### 3.3. Impact of PCV13 on Different Clinical Presentation of ser3 Infection in Italian Children under 8

The number of cases of ser3 IPDs grouped for infection type (pneumonia, sepsis, or meningitis) in the different study groups is reported in Table 2.

The ASRs of ser3 pneumonia in children born before 2011 (group A), born after 2011 (group B), and in children born after 2011 and vaccinated (group B1) are reported in Figure 2B. The crude incidence rate of ser3 pneumonias was 0.76 per 1,000,000 p-y in group A (95% CI: 0.60−0.92). ASR was 0.85 per 1,000,000 p-y in group B (95% CI: 0.70−1.00) and 0.62 per 1,000,000 p-y in subgroup B1 (95% CI: 0.38−0.86) (Table 1). The SIR for ser3 pneumonia in vaccinated children compared to unvaccinated children born before 2011 was 1.00 (95% CI: 1.00−1.00). PCV13 impact on ser3 pneumonia was −0.12 (95% CI: −1.10−0.40).

Age-specific ser3 sepsis and meningitis incidence rates in the three groups are reported in Figure 2C. Severe systemic infections (sepsis and meningitis) had a crude incidence rate of 0.24 cases per 1,000,000 p-y in children born before 2011 (95% CI: −0.04−0.52). ASR was 0.02 per 1,000,000 p-y in children under 8 born after 2011 (95% CI: −0.25−0.29) and 0.00 per 1,000,000 p-y in the group of vaccinated children (Table 1). SIR showed that children born post-PCV13 are 81% less likely to develop ser3 severe systemic infections than those born pre-PCV13 (SIR = 0.19; 95% CI −0.58−0.96). PCV13 impact on ser3 severe systemic infections was 0.92 (95% CI: −1.58−1.00).

All through the study, severe systemic infections were 10/43 (23.3%) in unvaccinated patients and 0/16 (0.0%) in vaccinated children, with a statistically significant difference (*p* = 0.0487) (Figure 4A; Table 3). Patients admitted to the pediatric intensive care unit (PICU) were 17/43 (39.5%) among non-vaccinated children and 1/16 (6.3%) in vaccinated children (*p* = 0.0233, OR = 0.10, 95% CI = 0.005−0.87) (Figure 4B; Table 3). PCV13-vaccinated children are thus less prone to develop ser3-IPDs with extremely severe clinical course than children who did not received PCV13 (*p* = 0.0233, OR = 0.10, 95% CI = 0.005−0.87).

## 4. Discussion

Our study shows for the first time that the ser3 vaccine has a different impact on different clinical presentations. In the post-PCV13 period, ser3 IPD showed an overall case reduction of 13% (impact: 0.13). The impact was much greater for sepsis and meningitis, with a case reduction of 92% (impact: 0.92) in those born after PCV13 introduction. Overall, the risk of contracting the disease is 25% lower in vaccinated children compared to children born before PCV13 was available (2011). Our data are in agreement with literature data on PCV13 effectiveness against ser3: 26% effectiveness in children aged 4 to 55 months [2]; 25.9% effectiveness in children aged 7 to 59 months [35]; 79.5% effectiveness in children aged 2 to 59 months [4]; 74% effectiveness in children aged 74 to 729 days [13]; 68% effectiveness in children under 5 years of age [3]; and 63.5% pooled vaccine effectiveness in pediatric age based on a recent systematic meta-analysis of the literature [14]. On the other hand, our study shows for the first time that the impact of vaccination is different among different clinical presentations, with the highest impact on the most severe clinical presentations (meningitis and sepsis a), where the risk is 81% lower (SIR = 0.19) in those born post-PCV13, and the case reduction in vaccinated children is 92%.

Several reasons have been addressed to explain PCV13’s reduced effectiveness against ser3 pneumonias compared to other serotypes, among them microbiological features of the pathogen, such as the thick polysaccharide capsule and release of free CPS, which can saturate antibodies and thus reduce opsonophagocytic killing and neutralization [10,11].

PCV13 has been licensed based on the non-inferiority immunogenicity criteria established for the 7-valent vaccine (PCV7) [36]. This criteria is an aggregate correlate of protection represented by serum anti-CPS antibody titers ≥0.35 µg/mL [37]. Afterwards, several post-licensure studies have demonstrated wide variability in the levels of protection against the different serotypes [2]. In particular, for ser3, a much higher IgG titer of 2.83 µg/mL should be obtained to grant protection [2]. In addition, some data show lower immunogenicity of PCV13 versus ser3 antigen, resulting in a lower antibody concentration in response to vaccination [38,39,40,41]. It is known that the correlate of protection may differ depending on whether the aim of vaccination is to prevent invasive infections, mucosal infection, or carrier state [42]. Moreover, within invasive infections, the antibody levels required for protection may vary depending on disease severity or the degree of systemic involvement [16,42]. To the best of our knowledge no study has been conducted to evaluate a differential impact of PCV13 on different clinical presentation and severity of IPDs.

In our study, the incidence of ser3 pneumonia evaluated with indirect age standardization did not decrease after PCV13 introduction, but the evaluation of impact showed a case raise of 12% post-PCV13. In this regard, it is necessary to underline the presence of a bias in the study. Actually, the number of samples received by central lab significantly increased—doubling over the study period—due to the introduction of molecular surveillance in the diagnoses of IPD and the improvement in the attention paid by clinicians to surveillance all over Italy (as also confirmed by the Italian National Institute of health; see Appendix A), so the possibility of recording an IPD doubled over the years. All samples sent to the central lab with the clinical suspicion of invasive bacterial disease were tested, amongst others, for the *lytA* gene, in order to minimize bias brought about by changes in the diagnostic practices of physicians, like that of assuming that the risk of IPD is much lower after PCV introduction. The above-mentioned bias has undoubtedly limited the significance of vaccination impact. Nevertheless, MSR is the only surveillance database in Italy recording vaccination status, so our data can be considered the most informative on a national scale. Interestingly, residual diseases due to ser3 in children aged 0–1, 1–2, and 2–3 years were not observed post-PCV13. When it comes to the evaluation of impact, this is of particular significance, especially considering that those age groups are the most affected by IPD among young children in Italy [43].

Different from pneumonias, severe systemic infections like sepsis and meningitis showed a consistent reduction after PCV13 introduction, even in the presence of the above-mentioned bias, with no registered cases among vaccinated patients.

Moreover, the proportion of systemic infections was significantly higher in unvaccinated versus vaccinated children (23.3% versus 0.0%) and systemic infections requiring intensive care support decreased from 39.5% in unvaccinated patients to 6.3% in vaccinated ones, with high statistical significance.

Our data analysis did not allow a comprehensive evaluation of secular trend but the stability in the incidence rates of overall ser3 IPD during the study period does not lean toward the presence of a secular trend for ser3. The trend in the reduction of overall IPD (Figure 3), despite the limited impact of PCV13 against ser3 pneumonias, can thus be ascribable to the significant reduction of the severe systemic infections which represent a small proportion of ser3 IPD.

The present study has some limitations due to the exiguity of the sample and the impossibility to determine real national incidences of IPD because of voluntary participation in MSR. However, to the best of our knowledge, this is the first study evaluating the impact of pneumococcal serotype 3 vaccination on different clinical presentations and suggesting a greater and significant impact on the most severe clinical presentations. Even though our results cannot be considered conclusive, we aim to open the way to assessments that stratify the impact of vaccination based on the severity of infections, going beyond the mere overall occurrence of IPD.

## 5. Conclusions

In conclusion, our study demonstrates that PCV13 is significantly effective against ser3 sepsis and meningitis, which disappeared in the post-PCV13 period, while its impact on ser3 pneumonias is limited, as previously demonstrated [2,3,4,5]. Larger studies should be conducted to evaluate the impact of PCV13 on ser3 IPD, taking into account the clinical severity of pneumococcal disease. While waiting for increasingly effective anti-pneumococcal vaccines targeting the highest possible number of serotypes, a vaccine such as PCV13, which guarantees protection against the most severe systemic infections caused by ser3, is currently the most effective prevention option available against this serotype.

## Figures and Tables

**Figure 1 vaccines-07-00128-f001:**
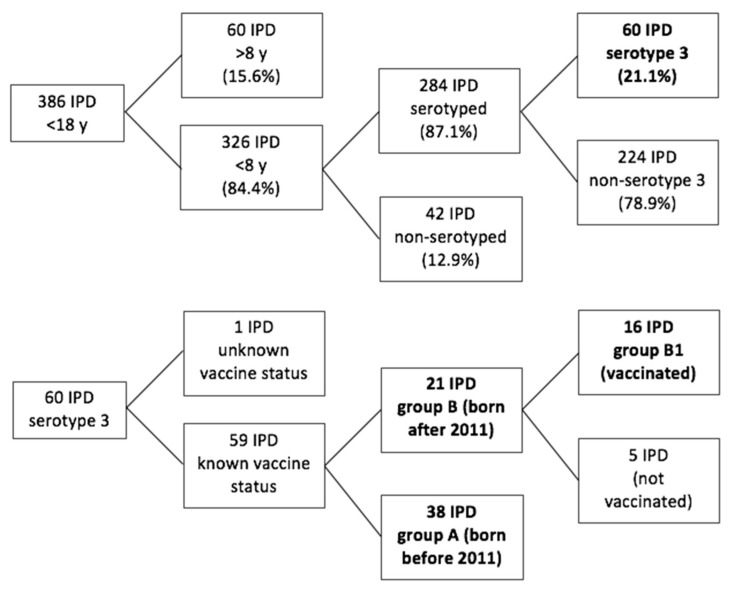
Study design showing the subdivision of the study population and numeric representation of the different study groups. The most relevant study groups are presented in bold. IPD: Invasive pneumococcal disease; y: years of age.

**Figure 2 vaccines-07-00128-f002:**
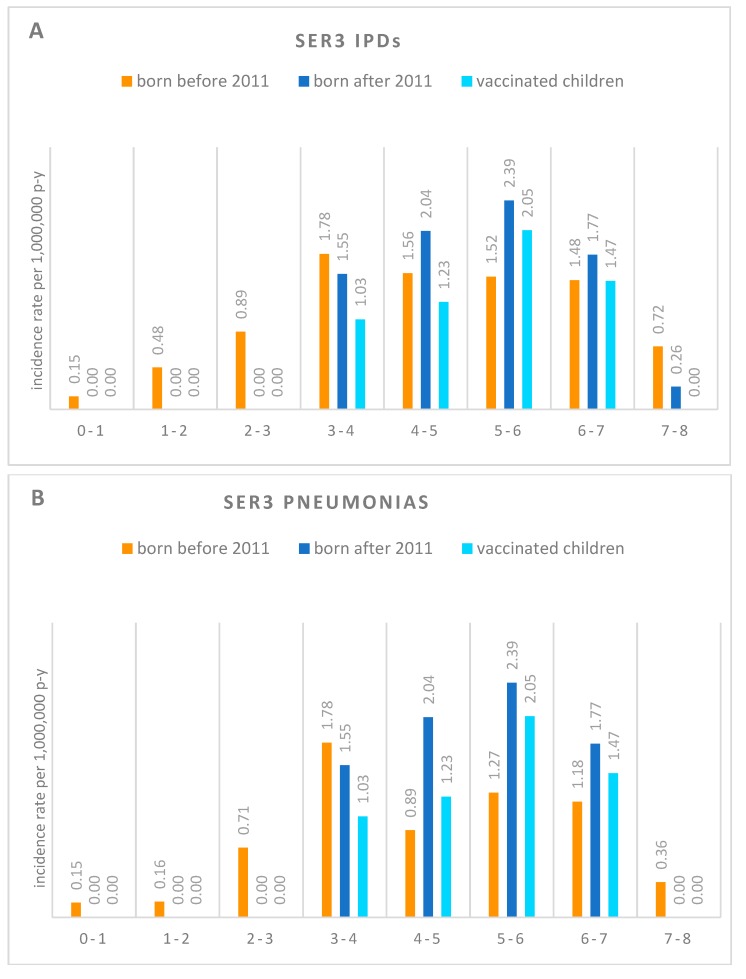
Age-specific incidence rates of serotype 3 *Streptococcus pneumoniae* (ser3) invasive pneumococcal disease (IPD) (**A**), pneumonias (**B**), and severe systemic infections (sepsis and meningitis) (**C**). Each subgroup refers to a one-year age class (ranging from 0–1 year old to 7–8 year old children) and shows incidence rates in children born before PCV13 introduction in the Italian vaccination calendar (2011), in children born after 2011, and in children born after 2011 and vaccinated. Age-specific rates were determined for intervals of one year of age (e.g., “3–4” stands for “patients with age ≥3 and <4 years of age”). P-y: person-years.

**Figure 3 vaccines-07-00128-f003:**
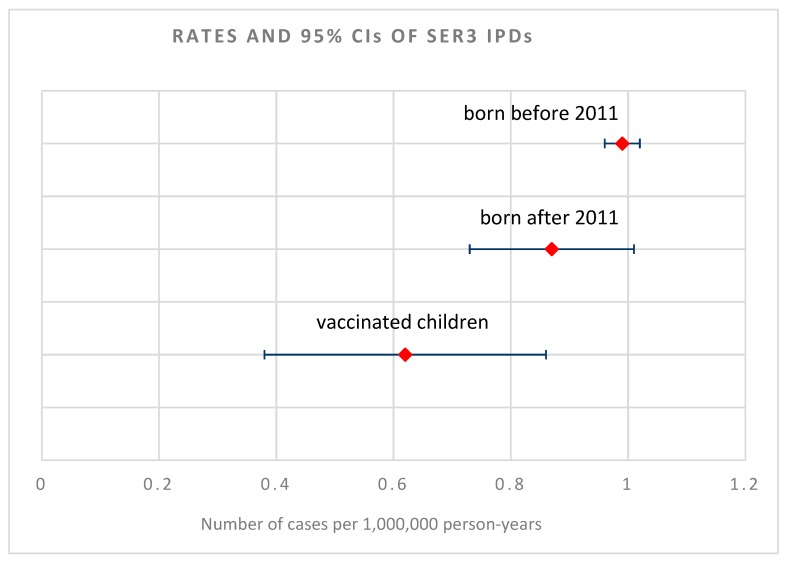
Incidence rates and 95% confidence intervals (CI) of serotype 3 (ser3) *Streptococcus pneumoniae* invasive disease (IPD) in children under 8 years of age born before PCV13 introduction in Italian vaccination calendar (2011), in children born after 2011 and in children born after 2011 and vaccinated with PCV13. Graph shows great overlap among CI and a reduction trend in ASR punctual values and CI from those born before 2011 to those born after 2011 coming to vaccinated children only. CIs are patently wider for the group of children born after 2011 and the group of vaccinated ones as adjusting rates for age, statistically augment variability.

**Figure 4 vaccines-07-00128-f004:**
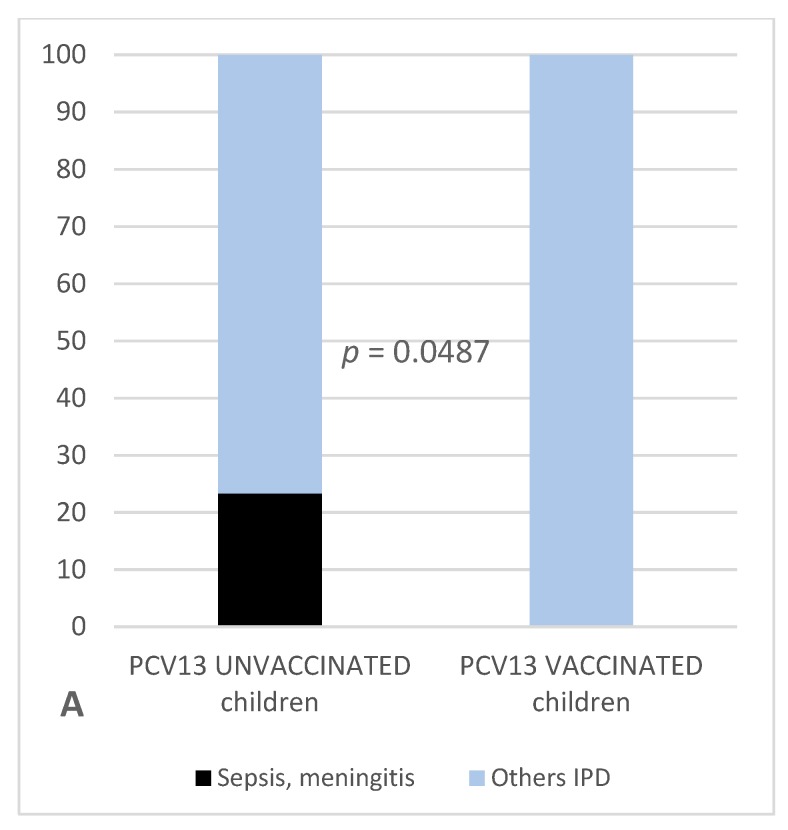
Percentage representation of the different forms of serotype 3 *Streptococcus pneumoniae* invasive pneumococcal disease (IPD) in children under 8 years of age, vaccinated with PCV13 and unvaccinated. Columns show internal group percentage subdivision based on clinical course severity: black represents sepsis and meningitis in panel **A**, with the addition of patients admitted to the pediatric intensive care unit (PICU) in panel **B**; other less severe forms of IPDs (different types of complicated pneumonias) are represented in light blue.

**Table 1 vaccines-07-00128-t001:** Incidence rates (first row), age standardized incidence rates (ASR) (second and third row), and 95% confidence intervals of serotype 3 (ser3) invasive pneumococcal disease (IPD), ser3 pneumonias, and ser3 severe systemic infections (sepsis and meningitis) in Italian children under 8 years of age born before 2011, after 2011, and vaccinated with PCV13. Incidence rates refer to 1,000,000 person-years. NaN: not available number.

Study Groups	Ser3 IPD	Ser3 Pneumonias	Ser3 Sepsis/Meningitis
**Children born before 2011**	0.99 (0.96–1.02)	0.76 (0.60–0.92)	0.24 (−0.04–0.52)
Crude incidence rates			
**Children born after 2011**	0.87 (0.73–1.01)	0.85 (0.70–1.01)	0.02 (−0.25–0.29)
Age-standardized incidence rates			
**Vaccinated children**	0.62 (0.38–0.86)	0.62 (0.38–0.86)	0.00 (NaN)
Age-standardized incidence rates			

**Table 2 vaccines-07-00128-t002:** Number of cases grouped for infection type in children under 8 years of age with serotype 3 *Streptococcus pneumoniae* (ser3) invasive pneumococcal infections born before and after PCV13 introduction in the Italian vaccination schedule (2011). Number of cases of ser3 IPD in PCV13-vaccinated children under 8 are also reported. Clinical subgroups are shown below the main groups that are presented in bold.

Children Born before PCV13 Introduction	Type of Infection	Children Born after PCV13 Introduction	Children Vaccinated with PCV13
**29**	**Complicated Pneumonias** - with effusion - with empyema - necrotizing/massive	**21**	**16**
11		11	8
17		9	8
1		1	0
**4**	**Sepsis**	**0**	**0**
**5**	**Meningitis**	**1**	**0**
**38**	**Total**	**22**	**16**

**Table 3 vaccines-07-00128-t003:** Number of cases grouped for infection type in children under 8 years of age with serotype 3 *Streptococcus pneumoniae* invasive pneumococcal infections and known vaccine status. PCV13 vaccinated children were all born after 2011, while the PCV13 unvaccinated children group was derived from grouping together children born before 2011 plus those born in PCV13 that had not completed their age-appropriate vaccination series at the moment of sample collection. Clinical subgroups are shown below the main groups that are presented in bold.

PCV13 Unvaccinated	Type of Infection	PCV13 Vaccinated
**33**	**Complicated Pneumonias** - with effusion - with empyema - necrotizing/massive	**16**
13		8
18		8
2		0
**4**	**Sepsis**	**0**
**6**	**Meningitis**	**0**
**43**	**Total**	**16**

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
