# Peer review of "Impact of the 13-Valent Pneumococcal Conjugate Vaccine on Severe Invasive Disease Caused by Serotype 3 Streptococcus Pneumoniae in Italian Children"

_vaccines, 2019, doi:10.3390/vaccines7040128_

Round 1

Reviewer 1 Report

In this retrospective study, Dr. Lodi and his colleagues investigated an impact of 13-valent pneumococcal conjugate vaccine (PCV13) on the invasive pneumococcal disease (IPD) caused by strains of Streptococcus pneumoniae serotype 3 in children <8 years of age and. For this, authors reviewed cases of IPD reported in Italy over the period 5 years prior and 8.5 years after PCV13 introduction into the NIP as recorded by the national Molecular Surveillance Register (MSR), the system based on the voluntary contribution of Italian hospitals.  

Although there was no clear evidence that vaccination with PCV13 had a significant impact on overall serotype 3 IPD in vaccinated children, authors conclude that, albeit the “results cannot be considered conclusive” (line 377), the data suggests that the vaccine was effective against most severe forms of IPD manifested by sepsis or meningitis (line 61).

In general, this study should be of interest for, vaccinologists, health-policymakers, infectious disease specialists, pediatricians and scientist in the field of pneumococcal research. However, the reviewer is not convinced that all underlying results are reported transparently enough for a reader to agree with such a conclusion.

Major comments:

Criteria for a case being classified as IPD are not clear in the study.

Authors report on IPD (line 114) being “defined as an infection confirmed by the isolation of S. pneumoniae from normally sterile body site”.  Isolation means culturing of live pneumococcus from a sample, yet some cases were diagnosed exclusively by a culture-independent method of RT PCR. Information on how many of IPDs were actually diagnosed by isolating live pneumococcus from a sample representing a sterile body site (blood, CSF, pleural fluid or else) needs to be provided. It needs to be clarified what “biological samples” (line 100) stands for.

Separation of pneumonia cases from cases of sepsis without pneumonia gives an impression that not in all instances of pneumonia presence of S. pneumoniae was detected in a sterile body site.  

It is unclear why septic pneumonia is considered by authors to represent a less severe manifestation of IPD compared with sepsis in the absence of pneumonia? Right now, only cases of meningitis (n=6) seem to represent a truly severe form of IPD.

Cases  among children <8 years of age (n=326 over period of 13 years and 7 months) constituted only  minute fraction (<3%) of all IPDs recorded by MSR over that period (range from 500+ in 2007 to 1400+ in either 2016 or 2017, app. 10.500 cases in total over 11 years in Figure S1). Authors need to provide information on the overall incidence of IPD per study year in children under the age of 8 years, as it is now presented for all ages IPD in Figure S1.

Although Figure S1 does support claim there was an improvement in surveillance over the study period and that it could indeed lead to underestimation of the impact PVCV13 had in children, an introduction of PCV13 into NIP could also lead to changes in diagnostic practices in sentinel centers: with physicians assuming that the risk of IPD is much lower in vaccinated children.  This could result in a reduced number of samples being tested from children in post-PCV13 implementation period and fewer number of cases being detected.   

Another scenario not considered by authors is an impact of temporal (secular) trends in the incidence of IPD caused by individual serotypes that have been already reported for disease by serotype 3, in the youngest children in particular (see Fenoll et al. Vaccine 2014 DOI: 10.1016/j.vaccine.2015.08.009 or Harboe et al. CID 2010; DOI: 10.1086/649872). Trends like that could be behind a decline in serotype 3 IPD independent of any impact of PCV13.  

The reviewer finds all three tables to be unclear. The legend to Table 1 seems to describe non-existing features (rows).

Minor points: 

Authors report on PCV13 being introduced into Italian NIP in 2011 for all children born after January 1st that but it is stated what schedule was followed. This information would be helpful for readers outside of Italy.

Line 318: what does “negative vaccination status” mean? 

Line 324-5: what are the numbers behind a case reduction by 13% reported here?

The manuscript needs thorough editing for better English.

For example, reviewer advises authors to consider replacing

“global” and “globally” with “overall”;

“basing” with “using “ or “based”;

(line 79) “versus” with “against”;

(line 83) “well known” with “assumed” or with “reported”;

(line 86) “proving” with “reporting”;

describing years before and after January 2011 as “periods” rather  than “eras”;

(line 264) “increased by” and not “increased of” (also, what was the number behind this increase by 102%?);

etc.

Reviewer 2 Report

The authors investigated the effectiveness of the 13-valent pneumococcal conjugate vaccine (PCV13) against IPDs due to serotype3. 

Data suggests significant reduction in meningitis and sepsis due to serotype 3.   

 However, manuscript can be further strengthened by adding few comments in discussion or introduction.

What were the predominant serotypes in Italian children in the pre-PCV13-era (2006- 2010) and in post-PCV13-era (2011-2018)? Was serotype 3 predominant? Line 156-159: Authors grouped children as PCV13 unvaccinated and PCV13 vaccinated. When the term ‘unvaccinated’ it can mislead some readers, add a comment about PCV7 coverage % in the PCV13 unvaccinated group (2006- 2010).  Figure 2, y- axis title is missing. Incidence rate per 1,000,000p-y should be included in y-axis and figure 2 legend.  What is the peak incidence age for IPDs in Italian children? 0-2 years or 2-4 years?  Interestingly, in Figure 2, panel A, residual diseases due to serotype 3 in children 0-1,1-2, 2-3 were not observed, authors should comment about this in discussion section.      To help readers to view the peak incidence and gradual decline of serotype 3 IPDs after PCV13 introduction. In figure 2 additional panel D, may include annual incidence rates/1,000,00 p-y (2006- 2018) of serotype 3 IPDs and/or annual reduction in serotype 3 IPDs case numbers (2006- 2018) in Italian children. 

Round 2

Reviewer 1 Report

Minor points:

1. It might be appropriate to clarify (Abstract line 53 and Introduction line 72), that the event censoring periods was  the introduction of PCV13, hence pre-PCV13 period and post-PCV13 introduction period in the study.

2. Second sentence of the second paragraph of the Results (lines 218-220) could authors report her not only on percentage but also on absolute numbers of patients?

3. Top section of Figure 1: 80 and 286 does not sum up to 386; 284 and 42 does not sum up to 286.

4. Line 235: "Changes in..." instead of "modifications of..."?

5. Legend to Table 1: Raw or row?

6. Line 395: "Among young children" instead of "in pediatric age"?

Author Response

1. It might be appropriate to clarify (Abstract line 53 and Introduction line 72), that the event censoring periods was  the introduction of PCV13, hence pre-PCV13 period and post-PCV13 introduction period in the study.

Response 1: The word "introduction" has been added as requested (line 53 and line 72).

2. Second sentence of the second paragraph of the Results (lines 218-220) could authors report her not only on percentage but also on absolute numbers of patients?

Response 2: Absolute numbers are now reported as requested (lines 222-223).

3. Top section of Figure 1: 80 and 286 does not sum up to 386; 284 and 42 does not sum up to 286.

Response 3: We are grateful for noticing this unpleasant typing error. Correction made (line 230).

4. Line 235: "Changes in..." instead of "modifications of..."?

Response 4: The change was applied (line 235).

5. Legend to Table 1: Raw or row?

Response 5: Row. Correction made (line 273).

6. Line 395: "Among young children" instead of "in pediatric age"?

Response 6: The change was applied (line 396).

Reviewer 2 Report

I have no further comments. 

Author Response

I have no further comments. 

We are grateful for the attention paid to our work.